# Learning Scalar Fields for Molecular Docking with Fast Fourier Transforms

**Bowen Jing, Tommi Jaakkola, Bonnie Berger**
CSAIL, Massachusetts Institute of Technology
bjing@mit.edu, {tommi, bab}@csail.mit.edu

## Abstract

Molecular docking is critical to structure-based virtual screening, yet the throughput of such workflows is limited by the expensive optimization of scoring functions involved in most docking algorithms. We explore how machine learning can accelerate this process by learning a scoring function with a functional form that allows for more rapid optimization. Specifically, we define the scoring function to be the cross-correlation of multi-channel ligand and protein scalar fields parameterized by equivariant graph neural networks, enabling rapid optimization over rigid-body degrees of freedom with fast Fourier transforms. Moreover, the runtime of our approach can be amortized at several levels of abstraction, and is particularly favorable for virtual screening settings with a common binding pocket. We benchmark our scoring functions on two simplified docking-related tasks: decoy pose scoring and rigid conformer docking. Our method attains similar but faster performance on crystal structures compared to the Vina and Gnina scoring functions, and is more robust on computationally predicted structures.

## 1 Introduction

Proteins are the macromolecular machines that drive almost all biological processes, and much of early-stage drug discovery focuses on finding molecules which bind to and modulate their activity. *Molecular docking*—the computational task of predicting the binding pose of a small molecule to a protein target—is an important step in this pipeline. Traditionally, docking has been formulated as an optimization problem over a *scoring function* designed to be a computational proxy for the free energy (Torres et al., 2019; Fan et al., 2019). Such scoring functions are typically a sum of pairwise interaction terms between atoms with physically-inspired functional forms (Quiroga & Villarreal, 2016). While these terms are simple and hence fast to evaluate, exhaustive sampling or optimization over the space of ligand poses is difficult and leads to the significant runtime of docking software.

ML-based scoring functions for docking have been an active area of research, ranging in sophistication from random forests to deep neural networks (Yang et al., 2022; Crampon et al., 2022). These efforts have largely sought to more accurately model the free energy based on a docked pose, which is important for downstream identification of binders versus non-binders (*virtual screening*). However, they have not addressed nor reduced the computational cost required to produce these poses in the first place. Hence, independently of the accuracy of these workflows, molecular docking for large-scale structure-based virtual screening remains computationally challenging, especially with the growing availability of large billion-compound databases such as ZINC (Tingle et al., 2023).

In this work, we explore a different paradigm and motivation for machine learning scoring functions, with the specific aim of *accelerating scoring and optimization* of ligand poses for high-throughput molecular docking. To do so, we forego the physics-inspired functional form of commonly used scoring functions, and instead frame the problem as that of learning *scalar fields* independently associated with the 3D structure of the protein and ligand, respectively. We then define the score to

NeurIPS 2023 AI for Science Workshop.

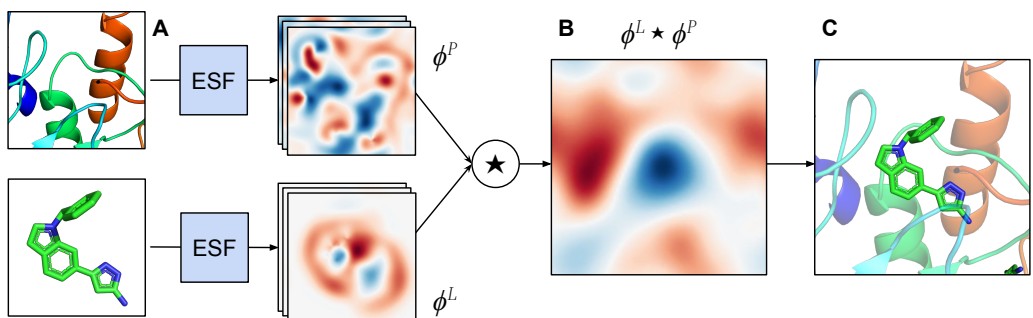

Figure 1: **Overview** of the scalar field-based scoring function and docking procedure. The translational FFT procedure is shown here; the rotational FFT is similar, albeit harder to visualize. (A) The protein pocket and ligand conformer are independently passed through equivariant scalar field networks (ESFs) to produce scalar fields. (B) The fields are cross-correlated to produce heatmaps over ligand translations. (C) The ligand coordinates are translated to the argmax of the heatmap. Additional scalar field visualizations are in Appendix D.

be the cross-correlation between the overlapping scalar fields when oriented according to the ligand pose. While seemingly more complex than existing scoring functions, these cross-correlations can be rapidly evaluated over a large number of ligand poses simultaneously using Fast Fourier Transforms (FFT) over both the translational space $\mathbb{R}^3$ and the rotational space $SO(3)$. This property allows for significant speedups in the optimization over these degrees of freedom.

Further contrasting with existing ML scoring functions, the computational cost of our method can be *amortized* at several levels of abstraction, significantly accelerating runtimes for optimized workflows. For example, unlike methods that require one neural network forward pass per pose, our network is evaluated once per protein structure or ligand conformer *independently*. Post-amortization, we attain translational and rotational optimization runtimes as fast as 160 $\mu$s and 650 $\mu$s, respectively, with FFTs. Such throughputs, when combined with effective sampling and optimization, could make docking of very large compound libraries feasible with only modest resources.

Empirically, we evaluate our method on two simplified docking-related tasks: (1) decoy pose scoring and (2) rigid conformer docking. On both tasks, our scoring function is competitive with—but faster than—the scoring functions of Gnina (Ragoza et al., 2017; McNutt et al., 2021) and Vina (Trott & Olson, 2010) on PDBBind crystal structures and is significantly better on ESMFold structures. We then demonstrate the further advantages of runtime amortization on the virtual screening-like setup of the PDE10A test set (Tosstorff et al., 2022), where—since there is only one unique protein structure—our method obtains a 50x speedup in total inference time at no loss of accuracy.

## 2 Method

### 2.1 Equivariant Scalar Fields

We consider the inputs to a molecular docking problem to be a pair of protein structure and ligand molecule, encoded as a featurized graphs $G^P, G^L$, and with the protein structure associated with alpha carbon coordinates $\mathbf{X}^P = [\mathbf{x}_1^P, \dots \mathbf{x}_{N_P}^P] \in \mathbb{R}^{3 \times N_P}$. The molecular docking problem is to find the ligand atomic coordinates $\mathbf{X}^L = [\mathbf{x}_1^L, \dots \mathbf{x}_{N_L}^L] \in \mathbb{R}^{3 \times N_L}$ of the true binding pose. To this end, our aim is to parameterize and learn (multi-channel) scalar fields $\phi^P := \phi(\mathbf{x}; G^P, \mathbf{X}^P)$ and $\phi^L := \phi^L(\mathbf{x}; G^L, \mathbf{X}^L)$ associated with the protein and ligand structures, respectively, such that the scoring function evaluated on any pose $\mathbf{X}^L \in \mathbb{R}^{3N_L}$ is given by

$$E(\mathbf{X}^P, \mathbf{X}^L) = \sum_c \int_{\mathbb{R}^3} \phi_c^P(\mathbf{x}; G^P, \mathbf{X}^P)\phi_c^L(\mathbf{x}; G^L, \mathbf{X}^L)\, d^3\mathbf{x} \tag{1}$$

where $\phi_c$ refers to the $c^{\text{th}}$ channel of the scalar field. While neural fields that directly learn functions $\mathbb{R}^3 \to \mathbb{R}$ have been previously developed as encodings of molecular structures (Zhong et al., 2019), such a formulation is unsuitable here as the field must be defined relative to the variable-sized structure graphs $G^P, G^L$ and transform appropriately with rigid-body motions of their coordinates.

Instead, we propose to parameterize the scalar field as a sum of contributions from each ligand atom or protein alpha-carbon, where each contribution is defined by its coefficients in a *spherical harmonic expansion* centered at that atom (or alpha-carbon) coordinate in 3D space. To do so, we choose a set $R_j : \mathbb{R}^+ \to \mathbb{R}$ of radial basis functions (e.g., Gaussian RBFs) in 1D and let $Y_m^\ell$ be the real spherical harmonics. Then we define

$$\phi_c(\mathbf{x}; G, \mathbf{X}) = \sum_{n,j,\ell,m} A_{cnj\ell m}(G, \mathbf{X}) R_j(\|\mathbf{x} - \mathbf{x}_n\|) Y_\ell^m \left( \frac{\mathbf{x} - \mathbf{x}_n}{\|\mathbf{x} - \mathbf{x}_n\|} \right) \tag{2}$$

where here (and elsewhere) we drop the superscripts $L, P$ for common definitions. Given some constraints on how the vector of coefficients $A_{cnj\ell m}$ transforms under $SE(3)$, this parameterization of the scalar field satisfies the following important properties:

**Proposition 1.** *Suppose the scoring function is parameterized as in Equation 2 and for any $R \in SO(3), \mathbf{t} \in \mathbb{R}^3$ we have $A_{cnj\ell m}(G, R.\mathbf{X} + \mathbf{t}) = \sum_{m'} D_{mm'}^\ell(R) A_{cnj\ell m'}(G, \mathbf{X})$ where $D^\ell(R)$ are the (real) Wigner D-matrices, i.e., irreducible representations of $SO(3)$. Then for any $g \in SE(3)$,*

1. *The scalar field transforms equivariantly: $\phi_c(\mathbf{x}; G, g.\mathbf{X}) = \phi_c(g^{-1}.\mathbf{x}; G, \mathbf{X})$.*

2. *The scoring function is invariant: $E(g.\mathbf{X}^P, g.\mathbf{X}^L) = E(\mathbf{X}^P, \mathbf{X}^L)$.*

See Appendix B for the proof. We choose to parameterize $A_{cnj\ell m}(G, R.\mathbf{X})$ with E3NN graph neural networks (Thomas et al., 2018; Geiger & Smidt, 2022), which are specifically designed to satisfy these equivariance properties and produce all coefficients in a single forward pass. The core of our method consists of the training of two such equivariant scalar field networks (ESFs), one for the ligand and one for the protein, which then parameterize their respective scalar fields. While the second property (invariance of the scoring function) is technically the only one required by the problem symmetries, the first property ensures that different ligand poses related by rigid-body transformations can be evaluated via transformations of the scalar field itself (without re-evaluating the neural network) and is thus essential to our method.

Next, we show how this parameterization enables ligand poses related by rigid body motions to some reference pose to be rapidly evaluated with fast Fourier transforms (all derivations in Appendix B). There are actually two ways to do so: we can evaluate the score of all poses generated by *translations* of the reference pose, or via *rotations* around some fixed point (which we always choose to be the center of mass of ligand). These correspond to FFTs over $\mathbb{R}^3$ and $SO(3)$, respectively.

## 2.2 FFT Over Translations

We first consider the space of poses generated by translations. Given some reference pose $\mathbf{X}^L$, the score as a function of the translation is just the cross-correlation of the fields $\phi^L$ and $\phi^P$:

$$E(\mathbf{X}^P, \mathbf{X}^L + \mathbf{t}) = \sum_c \int_{\mathbb{R}^3} \phi_c^P(\mathbf{x}) \phi_c^L(\mathbf{x} - \mathbf{t}) \, d^3\mathbf{x} = \sum_c (\phi_c^L \star \phi_c^P)(\mathbf{t}) \tag{3}$$

where we have dropped the dependence on $G, \mathbf{X}$ for cleaner notation and applied Proposition 1. By the convolution theorem, these cross-correlations may be evaluated using Fourier transforms:

$$\phi_c^L \star \phi_c^P = \frac{1}{(2\pi)^{3/2}} \mathcal{F}^{-1} \left\{ \overline{\mathcal{F}[\phi_c^L]} \cdot \mathcal{F}[\phi_c^P] \right\} \tag{4}$$

Hence, in order to simultaneously evaluate all possible translations of the reference pose, we need to compute the Fourier transforms of the protein and ligand scalar fields. One naive way of doing so would be to explicitly evaluate Equation 2 at an evenly-spaced grid of points spanning the structure and then apply a fast Fourier transform. However, this would be too costly, especially during training time. Instead, we observe that the functional form allows us to immediately obtain the Fourier transform via the expansion coefficients $A_{cnj\ell m}$:

$$\mathcal{F}[\phi_c](\mathbf{k}) = \sum_n e^{-i\mathbf{k}\cdot\mathbf{x}_n} \sum_\ell (-i)^\ell \sum_{m,n} A_{cnj\ell m} \mathcal{H}_\ell[R_j](\|\mathbf{k}\|) Y_\ell^m(\mathbf{k}/\|\mathbf{k}\|) \tag{5}$$

where now $Y_\ell^m$ must refer to the complex spherical harmonics and the coefficients must be transformed correspondingly, and

$$\mathcal{H}_\ell[R_j](k) = \sqrt{\frac{2}{\pi}} \int_0^\infty j_\ell(kr) R_j(r) r^2 \, dr \tag{6}$$

is the $\ell^{\text{th}}$ order spherical Bessel transform of the radial basis functions. Importantly, $\mathcal{H}_\ell[R_j]$ and $Y_\ell^m$ can be precomputed and cached at a grid of points *independently* of any specific structure, such that only the translation terms and expansion coefficients need to be computed for every new example.

## 2.3 FFT Over Rotations

We next consider the space of poses generated by rotations. Suppose that given some reference pose $\mathbf{X}^L$, the protein and ligand scalar fields are both expanded around some common coordinate system origin using the complex spherical harmonics and a set of *global radial basis functions* $S_j(r)$:

$$\phi_c(\mathbf{x}) = \sum_{j,\ell,m} B_{cj\ell m} S_j(\|\mathbf{x}\|) Y_\ell^m(\mathbf{x}/\|\mathbf{x}\|) \tag{7}$$

We seek to simultaneously evaluate the score of poses generated via rigid rotations of the ligand, which (thanks again to Proposition 1) is given by the rotational cross-correlation

$$E(\mathbf{X}^P, R.\mathbf{X}^L) = \sum_c \int_{\mathbb{R}^3} \phi_c^P(\mathbf{x}) \phi_c^L(R^{-1}\mathbf{x}) \, d^3\mathbf{x} \tag{8}$$

Cross-correlations of this form have been previously studied for rapid alignment of crystallographic densities (Kovacs & Wriggers, 2002) and of signals on the sphere in astrophysics (Wandelt & Górski, 2001). It turns out that they can also be evaluated in terms of Fourier sums:

$$\int_{\mathbb{R}^3} \phi_c^P(\mathbf{x}) \phi_c^L(R^{-1}\mathbf{x}) \, d^3\mathbf{x} = \sum_{\ell,m,h,n} d_{mh}^\ell d_{hn}^\ell I_{mn}^\ell e^{i(m\xi + h\eta + n\omega)} \tag{9}$$

where $\xi, \eta, \omega$ are related to the the Euler angles of the rotation $R$, $d^\ell$ is the (constant) Wigner $D$-matrix for a rotation of $\pi/2$ around the $y$-axis, and

$$I_{mn}^\ell = \sum_{j,k} B_{cj\ell m}^P \overline{B_{ck\ell n}^L} G_{jk} \quad \text{where} \quad G_{jk} = \int_0^\infty S_j(r) S_k(r) r^2 \, dr \tag{10}$$

Thus the main task is to compute the complex coefficients $B_{cj\ell m}$ of the ligand and protein scalar fields, respectively. This is not immediate as the fields are defined using expansions in "local" radial and spherical harmonic bases, i.e., with respect to the individual atom positions as opposed to the coordinate system origin. Furthermore, since we cannot (in practice) use a complete set of radial or angular basis functions, it is generally not possible to express the ligand or protein scalar field as defined in Equation 2 using the form in Equation 7. Instead, we propose to find the coefficients $B_{cj\ell m}$ that give the best approximation to the true scalar fields, in the sense of least squared error.

Specifically, suppose that $\mathbf{R} \in \mathbb{R}^{N_{\text{grid}} \times N_{\text{local}}}$ are the values of $N_{\text{local}}$ real local basis functions (i.e., different origins, RBFs, and spherical harmonics) evaluated at $N_{\text{grid}}$ grid points and $\mathbf{A} \in \mathbb{R}^{N_{\text{local}}}$ is the vector of coefficients defining the scalar field $\phi_c$. Similarly define $\mathbf{S} \in \mathbb{R}^{N_{\text{grid}} \times N_{\text{global}}}$ using the real versions of the global basis functions. We seek to find the least-squares solution $\mathbf{B} \in \mathbb{R}^{N_{\text{global}}}$ to the overdetermined system of equations $\mathbf{RA} = \mathbf{SB}$, which is given by

$$\mathbf{B} = (\mathbf{S}^T\mathbf{S})^{-1}\mathbf{S}^T\mathbf{R}\mathbf{A} \tag{11}$$

Notably, this is simply a linear transformation of the local coefficients $A_{cnj\ell m}$. Thus, if we can precompute the inverse Gram matrix of the global bases $(\mathbf{S}^T\mathbf{S})^{-1}$ and the inner product of the global and local bases $\mathbf{S}^T\mathbf{R}$, then for any new scalar field $\phi_c$ the real global coefficients are immediately available via a linear transformation. The desired complex coefficients can then be easily obtained via a change of bases. At first glance, this still appears challenging due to the continuous space of possible atomic or alpha-carbon positions, but an appropriate discretization makes the precomputation relatively inexpensive without a significant loss of fidelity.

## 2.4 Training and Inference

We now study how the rapid cross-correlation procedures presented thus far are used in training and inference. For a given training example with protein structure $\mathbf{X}^P$, the scoring function $E(\mathbf{X}^P, \mathbf{X}^L)$ should ideally attain a maximum at the true ligand pose $\mathbf{X}^L = \mathbf{X}^{L\star}$. We equate this task to that of learning an *energy based model* to maximize the log-likelihood of the true pose under the model likelihood $p(\mathbf{X}^L) \propto \exp\left[E(\mathbf{X}^P, \mathbf{X}^L)\right]$. However, as is typically the case for energy-based models, directly optimizing this objective is difficult due to the intractable partition function.

Instead, following Corso et al. (2023), we conceptually decompose the ligand pose $\mathbf{X}^L$ to be a tuple $\mathbf{X}^L = (\mathbf{X}^C, R, \mathbf{t})$ consisting of a zero-mean conformer $\mathbf{X}^C$, a rotation $R$, and a translation $\mathbf{t}$, from which the pose coordinates are obtained: $\mathbf{X}^L = R.\mathbf{X}^C + \mathbf{t}$. Then consider the following *conditional* log-likelihoods:

$$\log p(\mathbf{t} \mid \mathbf{X}^C, R) = E(\mathbf{X}^P, \mathbf{X}^L) - \log \int_{\mathbb{R}^3} \exp\left[E(\mathbf{X}^P, R.\mathbf{X}^C + \mathbf{t}')\right] d^3\mathbf{t}' \tag{12a}$$

$$\log p(R \mid \mathbf{X}^C, \mathbf{t}) = E(\mathbf{X}^P, \mathbf{X}^L) - \log \int_{SO(3)} \exp\left[E\left(\mathbf{X}^P - \mathbf{t}, R'.\mathbf{X}^C\right)\right] dR' \tag{12b}$$

We observe that these integrands are precisely the cross-correlations in Equations 3 and 8, respectively, and can be quickly evaluated and summed for all values of $\mathbf{t}'$ and $R'$ using fast Fourier transforms. Thus, the integrals—which are the marginal likelihoods $p(\mathbf{X}^C, R)$ and $p(\mathbf{X}^C, \mathbf{t})$—are tractable and the conditional log-likelihoods can be directly optimized in order to train the neural network. Although neither technically corresponds to the joint log-likelihood of the pose, we find that these training objectives work well in practice and optimize their sum in our training procedure.

At inference time, a rigid protein structure $\mathbf{X}^P$ is given and the high-level task is to score or optimize candidate ligand poses $\mathbf{X}^L$. A large variety of possible workflows can be imagined; however, for proof of concept and for our experiments in Section 3 we describe and focus on the following relatively simple inference workflows (presented in greater detail in Appendix C):

- **Translational FFT (TF)**. Given a conformer $\mathbf{X}^C$, we conduct a grid-based search over $R$ and use FFT to optimize $\mathbf{t}$ in order to find the best pose $(\mathbf{X}^C, R, \mathbf{t})$. To do so, we compute the Fourier coefficients (Equation 5) of the protein $\mathbf{X}^P$ *once* and for *each* possible ligand orientation $R.\mathbf{X}^C$. We then use translational cross-correlations (Equation 3) to find the best translation $\mathbf{t}$ for each $R$ and return the highest scoring combination.

- **Rotational FFT (RF)**. Given a conformer $\mathbf{X}^C$, we conduct a grid-based search over $\mathbf{t}$ and use FFT to optimize $R$. To do so, we compute the global expansion coefficients $B^P_{cj\ell m}$ of the protein $\mathbf{X}^L - \mathbf{t}$ relative to *each* possible ligand position $\mathbf{t}$ and *once* for the ligand $\mathbf{X}^C$ relative to its (zero) center of mass (Equation 11). We then use rotational cross-correlations (Equation 8) to find the best orientation $R$ for each $\mathbf{t}$ and return the highest scoring combination.

- **Translational scoring (TS)**. Here we instead are given a list of poses $(\mathbf{X}^C, R, \mathbf{t})$ and wish to score them. Because the values of $R$ nor $\mathbf{t}$ may not satisfy a grid structure, we cannot use the FFT methods. Nevertheless, we can compute the (translational) Fourier coefficients of the protein $\mathbf{X}^P$ and for each unique oriented conformer $R.\mathbf{X}^C$ of the ligand using Equation 5. We then evaluate

$$E(\mathbf{X}^P, R.\mathbf{X}^C + \mathbf{t}) = \sum_c \int_{\mathbb{R}^3} \overline{\mathcal{F}[\phi_c^P](\mathbf{k})} \cdot \mathcal{F}[\phi_c^L(\,\cdot\,; R.\mathbf{X}^C)](\mathbf{k}) \cdot e^{-i\mathbf{k}\cdot\mathbf{t}} \, d^3\mathbf{k} \tag{13}$$

  Since the Fourier transform is an orthogonal operator on functional space, this is equal to the real-space cross-correlation.

- **Rotational scoring (RS)**. Analogously, we can score a list of poses $(\mathbf{X}^C, R, \mathbf{t})$ using the global spherical expansions $B_{cj\ell m}$. We obtain the real expansion coefficients of the protein relative to each $\mathbf{t}$ and for each ligand conformer $\mathbf{X}^C$ using Equation 11. The score for $(\mathbf{X}^C, R, \mathbf{t})$ is then given by the rotational cross-correlation

$$E(\mathbf{X}^P, R.\mathbf{X}^C + \mathbf{t}) = \sum_{c,j,k,\ell,m,n} B^P_{cj\ell m}(\mathbf{X}^P - \mathbf{t}) B^L_{ck\ell n}(\mathbf{X}^C) D^\ell_{mn}(R) G_{jk} \tag{14}$$

  where $G_{jk}$ is as defined in Equation 10 and $D^\ell_{mn}$ are the real Wigner $D$-matrices.

Table 1: **Typical runtimes** of the computations involved in inference-time scoring and optimization procedures, measured on PDBBind with one V100 GPU. The three sets of rows delineate computations that are protein-specific, ligand-specific, or involve both protein and ligand, respectively.

| Frequency | Computation | TF | RF | TS | RS | Runtime |
|---|---|---|---|---|---|---|
| Per protein structure | Coefficients $A_{cnj\ell m}$ | ✓ | ✓ | ✓ | ✓ | 65 ms |
|  | FFT coefficients | ✓ |  | ✓ |  | 7.0 ms |
| ↪ Per translation | Global expansion $B_{cj\ell m}$ |  | ✓ |  | ✓ | 80 ms |
| Per ligand conformer | Coefficients $A_{cnj\ell m}$ | ✓ | ✓ | ✓ | ✓ | 4.3 ms |
|  | Global expansion $B_{cj\ell m}$ |  | ✓ |  | ✓ | 17 ms |
| ↪ Per rotation | FFT coefficients | ✓ |  | ✓ |  | 1.6 ms |
| Per conformer × rotation | Translational FFT | ✓ |  |  |  | 160 $\mu$s |
| Per conformer × translation | Rotational FFT |  | ✓ |  |  | 650 $\mu$s |
| Per pose | Translational scoring |  |  | ✓ |  | 1.0 $\mu$s |
|  | Rotational scoring |  |  |  | ✓ | 8.2 $\mu$s |

The runtime of these workflows can vary significantly depending on the parameters, i.e., number of proteins, ligands, conformers, rotations, and translations, with amortizations possible at several levels. Table 1 provides a summary of the computations in each workflow, their frequencies, and typical runtimes. We highlight that the **RF** workflow is well-suited for virtual screening since the precomputations for the protein and ligand translations within a pocket can be amortized across all ligands. Furthermore, if the ligands are drawn from a shared library, their coefficients can also be precomputed independent of any protein, leaving only the rotational FFT as the cost per ligand-protein pair. Thus our method can lend itself to the engineering of very high-throughput workflows.

## 3 Experiments

We train and test our model on the PDBBind dataset (Liu et al., 2017) with splits as defined by Stärk et al. (2022) providing 16379, 968, and 363 train, validation, and test complexes, respectively. We train two variants of our model: ESF and ESF-N, where the latter is trained with rotational and translational noise injected into the examples to increase model robustness. In both, the protein network operates on all heavy atom nodes, but only the alpha-carbons contribute to the scalar field. The input features and message-passing layers are otherwise similar to Corso et al. (2023), except without ESM features. Hyperparameters are detailed in Appendix E. For evaluation, we consider both the co-crystal structures in the PDBBind test split and their counterpart ESMFold complexes as prepared by Corso et al. (2023). We also collect a test set of 77 crystal structures (none of which are in PDBBind) of phosphodiesterase 10A (PDE10A) with different ligands bound to the same pocket (Tosstorff et al., 2022). This industrially-sourced dataset is representative of a real-world use case for molecular docking and benchmarks the benefits of runtime amortization with our approach.

To evaluate our method against baselines, we note that a scoring function by itself is not directly comparable to complete docking programs, which also include tightly integrated conformer search, pose clustering, and local refinement algorithms. Here, however, we focus on the development of the *scoring function* itself independently of these other components. Thus, we consider two simplified settings for evaluating our model: (1) **scoring decoy poses** with the aim of identifying the best pose among them, and (2) **docking rigid conformers** to a given pocket, similar to the re-docking setup in Stärk et al. (2022). The first setting focuses on evaluating only the quality of the scoring function itself, whereas the second is a simplified version of a typical docking setting that circumvents some of the confounding factors while still allowing the benchmarking of FFT-accelerated optimization.

We select Gnina (McNutt et al., 2021) as the baseline docking software, which runs parallel MCMC chains to collect pose candidates that are then refined and re-ranked to produce the final prediction. For the scoring function, we evaluate Gnina's namesake CNN (Ragoza et al., 2017) as the state-of-the-art ML scoring function, as well as the traditional scoring function of Vina (Trott & Olson, 2010), one of the most well-established docking programs in the development of the field. Both scoring functions are widely used and are natively supported by the Gnina program.

Table 2: **Decoy scoring results**. All RMSDs are heavy-atom symmetry aware. For ease of comparison, the best numbers from our method (ESF) are underlined if not bolded.

| Method | Crystal structures | | | | ESMFold structures | | | | Time per | |
| | <2 Å AUROC | Top RMSD | Top Rank | % <2 Å | <2 Å AUROC | Top RMSD | Top Rank | % <2 Å | Pose | Complex |
| --- | --- | --- | --- | --- | --- | --- | --- | --- | --- | --- |
| Vina | **0.93** | **0.54** | **2** | **91** | 0.86 | 2.43 | 419 | 43 | 3.4 ms | 110 s |
| Gnina | 0.90 | 0.59 | 3 | 83 | 0.84 | 2.19 | 1110 | 46 | 13.0 ms | 426 s |
| ESF-TS | 0.87 | 0.59 | 3 | 87 | 0.82 | **1.38** | 24 | **57** | 1.0 $\mu$s | 3.2 s |
| ESF-RS | 0.87 | 0.63 | 3 | 85 | 0.82 | 1.75 | **22** | 53 | 8.2 $\mu$s | 5.7 s |
| ESF-N-TS | 0.92 | 0.69 | 4 | 81 | **0.87** | 1.64 | **22** | 54 | 1.0 $\mu$s | 3.2 s |
| ESF-N-RS | 0.92 | 0.75 | 5 | 80 | **0.87** | 1.74 | 26 | 53 | 8.2 $\mu$s | 5.7 s |

## 3.1 Scoring Decoys

For each PDBBind test complex, we generate $32^3 - 1 = 32767$ decoy poses by sampling 31 translational, rotational, and torsional perturbations to the ground truth pose and considering all their possible combinations. On median, the RMSD of the closest decoy is 0.4 angstroms (Å), and 1.6% of all poses ($n = 526.5$) are below 2 Å RMSD (Appendix E). We then score all poses using the Vina and Gnina scoring functions and with our method in both **TS** (Equation 13) and **RS** (Equation 14) modes. The quality of each scoring function is evaluated with the AUROC when used as a <2Å RMSD classifier, the RMSD of the top-ranked pose (Top RMSD), the rank of the lowest-RMSD pose (Top Rank), and the fraction of complexes for which the identified pose is under 2 Å RMSD.

As shown in Table 2, our method is competitive with the Gnina and Vina scoring functions on crystal structures and better on ESMFold structures. This improved robustness is expected since the interaction terms in traditional scoring functions are primarily mediated by sidechain atoms, which are imperfectly predicted by ESMFold, whereas our scalar fields only indirectly depend on the sidechains via residue-level coefficients. The noise-augmented training obtains higher AUROC but is weaker in terms of identifying the best poses. Curiously, this is also true on ESMFold structures, where we expect robustness to be more important. Overall, **TS** is superior in performance to **RS**, likely due to the spatially coarser representation of the scalar fields in the global spherical harmonic expansion (i.e., Equation 7) relative to the grid-based Cartesian expansion.

In terms of runtime per pose, our method is faster than Vina by several orders of magnitude, with even greater acceleration compared to the neural network-based Gnina. The runtime improvement per complex is more tempered since the different proteins and ligand in every complex limit the opportunity for amortization. In fact, of the total runtime per complex in Table 2, only 1% (**TS**) to 5% (**RS**) is due to the pose scoring itself, with the rest due to preprocessing that must be done for every new protein and ligand independently. Hence, the total possible runtime improvement per complex is significantly greater for more suitable workflows.

## 3.2 Docking Conformers

We consider the task of pocket-level docking where all methods are given as input the ground-truth conformer in a random orientation. Following common practice (McNutt et al., 2021), we aim to provide 4 Å of translational uncertainty around the true ligand pose in order to define the binding pocket. To do so, we provide Gnina with a bounding box with 4 Å of padding around the true pose, and provide our method with a cube of side length 8 Å as the search space for $\mathbf{t}$ (with a random grid offset). For PDE10A, we define the pocket using the pose of the first listed complex (PDB 5SFS) and cross-dock to that protein structure. In all docking runs, we deactivate all torsion angles so that Gnina docks the provided conformer to the pocket. Default hyperparameters—and results for varying hyperparameters—are detailed in Appendix E and Appendix F, respectively.

As shown in Table 3, the baseline scoring functions obtain excellent performance on the PDBBind crystal structures, with a success rate of 79%. Our method is slightly weaker but also obtains high success rates (73%). The performance decrease in terms of Median RMSD is somewhat larger,

Table 3: **Rigid conformer docking results.** All RMSDs are heavy-atom symmetry aware. The median RMSD of our method (ESF) is lower-bounded at 0.5–0.6 Å by the resolution of the search grid (Appendix F). The runtime is shown as an average per complex, excluding / including pre-computations that can be amortized. The best numbers from ESF are underlined if not bolded.

| | PDBBind test | | | | | PDE10A | | |
| | Crystal | | ESMFold | | | | | |
| Method | % <2 Å | Med. RMSD | % <2 Å | Med. RMSD | Runtime | % <2 Å | Med. RMSD | Runtime |
|---|---|---|---|---|---|---|---|---|
| Vina | **79** | **0.32** | 24 | 6.1 | 20 s | **74** | **0.75** | 6.1 s |
| Gnina | 77 | 0.33 | 28 | 5.9 | 23 s | 73 | 0.77 | 6.0 s |
| ESF-TF | 70 | 1.13 | 31 | 4.6 | 0.8 s / 8.3 s | 67 | 1.20 | 1.0 s / 7.1 s |
| ESF-RF | 71 | 0.97 | 32 | 4.4 | 0.5 s / 67 s | 73 | 0.82 | 0.5 s / 1.5 s |
| ESF-N-TF | 72 | 1.10 | 46 | **2.9** | 0.7 s / 8.2 s | 64 | 1.11 | 1.0 s / 7.2 s |
| ESF-N-RF | 73 | 1.00 | **47** | 3.0 | 0.5 s / 68 s | 70 | 1.00 | 0.5 s / 1.5 s |

likely due to the coarse search grid over non-FFT degrees of freedom (Appendix F) and the lack of any refinement steps (which are an integral part of Gnina) in our pipeline. On ESMFold structures, however, our method obtains nearly twice the success rate (47% vs 28%) of the baseline scoring functions. Unlike in decoy scoring, noisy training noticeably contributes to the performance on ESMFold structures, and the **RF** procedure generally outperforms **TF**, likely due to the relatively finer effective search grid in rotational cross-correlations (Appendix F).

Because of the nature of the PDBBind workflow, the total runtime is comparable to or slower than the baselines when precomputations are taken into account. However, in terms of the pose optimization itself, our method is significantly faster than the Gnina baselines, despite performing a brute force search over the non-FFT degrees of freedom. While it is also possible to trade-off performance and runtime by changing various Gnina settings from their default values, our method expands the Pareto front currently available with the Gnina pipeline (Appendix F; Figure 6). This favorable tradeoff affirms the practical value-add of our method in the context of existing approaches.

To more concretely demonstrate the runtime improvements of our method with amortization, we then dock the conformers in the PDE10A dataset. Our method again has similar accuracy to the baselines (Table 3); however, because of the common pocket, all protein-level quantities are computed *only once* and the total runtime is significantly accelerated. For the **RF** procedure in particular, the computation of global coefficients on the translational grid is by far the most expensive step (Appendix F; Table 5), and the remaining ligand precomputations are very cheap. The amortization of these coefficients leads to a 45x speedup in the overall runtime (67 s → 1.5 s). (The runtime for Gnina is also accelerated, although to a lesser extent, due to the smaller ligand size.) As the number of ligands increases further, the total runtime per complex of our method would further decrease.

## 4 Conclusion

We have proposed a machine-learned based scoring function for accelerating pose optimization in molecular docking. Different from existing scoring functions, the score is defined as a cross-correlation between scalar fields, which enables the use of FFTs for rapid search and optimization. We have formulated a novel parameterization for such scalar fields with equivariant neural networks, as well as training and inference procedures with opportunities for significant runtime amortization. Our scoring function shows comparable performance but improved runtime on two simplified docking-related tasks relative to standard optimization procedures and scoring functions. Thus, our methodology holds promise when integrated with other components into a full docking pipeline. These integrations may include multi-resolution search, refinement with traditional scoring functions, and architectural adaptations for conformational (i.e., torsional) degrees of freedom—all potential directions of future work. In a broader context, we hope our work serves as a bridge between graph-based molecular machine learning and the literature on cross-correlations in computational structural biology and inspire related methods for other applications.

## Acknowledgments

We thank Hannes Stärk, Samuel Sledzieski, Ruochi Zhang, Michael Brocidiacono, Gabriele Corso, Xiang Fu, Felix Faltings, Ameya Daigavane, and Mario Geiger for helpful feedback and discussions. This work was supported by the NIH NIGMS under grant #1R35GM141861 and a Department of Energy Computational Science Graduate Fellowship.

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

# A Background

**Molecular docking.** The two key components of a molecular docking algorithm are (1) one or more scoring functions for ligand poses, and (2) a search, sampling, or optimization procedure. There is considerable variation in the design of these components and how they interact with each other, ranging from exhaustive enumeration and filtering (Shoichet et al., 1992; Meng et al., 1992) to genetic, gradient-based, or MCMC optimization algorithms (Trott & Olson, 2010; Morris et al., 1998; McNutt et al., 2021). We refer to reviews elsewhere (Ferreira et al., 2015; Torres et al., 2019; Fan et al., 2019) for comprehensive details. These algorithms have undergone decades of development and have been given rise to well-established software packages in academia and industry, such as AutoDock (Morris & Lim-Wilby, 2008), Vina (Trott & Olson, 2010) and Glide (Halgren et al., 2004). In many of these, the scoring function is designed not only to identify the binding pose, but also to predict the binding affinity or activity of the ligand (Su et al., 2018). In this work, however, we focus on learning and evaluating scoring functions for the rapid prediction of binding poses.

**ML methods in docking.** For over a decade, ML methods have been extensively explored to improve scoring functions for already-docked ligand poses, i.e., for prediction of activity and affinity in structural-based virtual screens (Li et al., 2021; Yang et al., 2022; Crampon et al., 2022). On the other hand, developing ML scoring functions as the direct optimization objective has required more care due the enormous number of function evaluations involved. MedusaNet (Jiang et al., 2020) and Gnina (Ragoza et al., 2017; McNutt et al., 2021) proposed to sparsely use CNNs for guidance and re-ranking (respectively) in combination with a traditional scoring function. DeepDock (Méndez-Lucio et al., 2021) used a hypernetwork to predict complex-specific parameters of a simple statistical potential. Recently, geometric deep learning models have explored entirely different paradigms for docking via direct prediction of the binding pose (Stärk et al., 2022; Zhang et al., 2022; Lu et al., 2022) or via a generative model over ligand poses (Corso et al., 2023).

**FFT methods in docking.** Methods based on fast Fourier transforms have been widely applied for the related problem of *protein-protein docking*. Katchalski-Katzir et al. (1992) first proposed using FFTs over the translational space $\mathbb{R}^3$ to rapidly evaluate poses using scalar fields that encode the shape complementarity of the two proteins. Later works extended this method to rotational degrees of freedom (Ritchie & Kemp, 2000; Ritchie et al., 2008; Padhorny et al., 2016) and additional scoring terms, such as pairwise electrostatic potentials and solvent accessibility (Gabb et al., 1997; Mandell et al., 2001; Chen & Weng, 2002). Today, FFT methods are a routine step in protein-protein docking programs such as PIPER (Kozakov et al., 2006), ClusPro (Kozakov et al., 2017), and HDOCK (Yan et al., 2020), where they enable the evaluation of billions of poses, typically as an initial screening step before further evaluation and refinement with a more accurate scoring function.

In contrast, FFT methods have been significantly less studied for protein-ligand docking. While a few works have explored this direction (Padhorny et al., 2018; Ding et al., 2020; Nguyen et al., 2018), these algorithms have not been widely adopted nor been incorporated into established docking software. A key limitation is that protein-ligand scoring functions are typically more complicated than protein-protein scoring functions and cannot be easily expressed as a cross-correlation between scalar fields (Ding et al., 2020). To our knowledge, no prior works have explored the possibility of overcoming this limitation by *learning* cross-correlation based scoring functions.

# B Mathematical Details

## B.1 Proof of Proposition 1

**Proposition 1.** *Suppose the scoring function is parameterized as in Equation 2 and for any $R \in SO(3), \mathbf{t} \in \mathbb{R}^3$ we have $A_{cnj\ell m}(G, R.\mathbf{X} + \mathbf{t}) = \sum_{m'} D^\ell_{mm'}(R) A_{cnj\ell m'}(G, \mathbf{X})$ where $D^\ell(R)$ are the (real) Wigner D-matrices, i.e., irreducible representations of $SO(3)$. Then for any $g \in SE(3)$,*

1. *The scalar field transforms equivariantly: $\phi_c(\mathbf{x}; G, g.\mathbf{X}) = \phi_c(g^{-1}.\mathbf{x}; G, \mathbf{X})$.*

2. *The scoring function is invariant: $E(g.\mathbf{X}^P, g.\mathbf{X}^L) = E(\mathbf{X}^P, \mathbf{X}^L)$.*

*Proof.* Let the action of $g = (R, \mathbf{t}) \in SE(3)$ be written as $g : \mathbf{x} \mapsto R\mathbf{x} + \mathbf{t}$ and hence $g^{-1} : \mathbf{x} \mapsto R^T(\mathbf{x} - \mathbf{t})$. We first note that $\|\mathbf{x} - g.\mathbf{x}_n\| = \|g^{-1}.\mathbf{x} - \mathbf{x}_n\|$ and $R^T(\mathbf{x} - g.\mathbf{x}_n) = g^{-1}.\mathbf{x} - \mathbf{x}_n$. Then

$$
\begin{aligned}
\phi_c(\mathbf{x}; G, g.\mathbf{X}) &= \sum_{n,j,\ell,m} A_{cnj\ell m}(G, R.\mathbf{X} + \mathbf{t}) R_j(\|\mathbf{x} - g.\mathbf{x}_n\|) Y_\ell^m\left(\frac{\mathbf{x} - g.\mathbf{x}_n}{\|\mathbf{x} - g.\mathbf{x}_n\|}\right) \\
&= \sum_{n,j,\ell,m'} A_{cnj\ell m'}(G, \mathbf{X}) R_j(\|\mathbf{x} - g.\mathbf{x}_n\|) \sum_m D_{mm'}^\ell(R) Y_\ell^m\left(\frac{\mathbf{x} - g.\mathbf{x}_n}{\|\mathbf{x} - g.\mathbf{x}_n\|}\right) \\
&= \sum_{n,j,\ell,m'} A_{cnj\ell m'}(G, \mathbf{X}) R_j(\|\mathbf{x} - g.\mathbf{x}_n\|) Y_\ell^{m'}\left(\frac{R^T(\mathbf{x} - g.\mathbf{x}_n)}{\|\mathbf{x} - g.\mathbf{x}_n\|}\right) \\
&= \sum_{n,j,\ell,m'} A_{cnj\ell m'}(G, \mathbf{X}) R_j(\|g^{-1}.\mathbf{x} - \mathbf{x}_n\|) Y_\ell^{m'}\left(\frac{g^{-1}.\mathbf{x} - \mathbf{x}_n}{\|g^{-1}.\mathbf{x} - \mathbf{x}_n\|}\right) \\
&= \phi_c(g^{-1}.\mathbf{x}; G, \mathbf{X})
\end{aligned}
$$

Next,

$$
\begin{aligned}
E(g.\mathbf{x}^P, g.\mathbf{x}^L) &= \sum_c \int_{\mathbb{R}^3} \phi_c^P(\mathbf{x}; G^P, g.\mathbf{X}^P) \phi_c^L(\mathbf{x}; G^L, g.\mathbf{X}^L) \, d^3\mathbf{x} \\
&= \sum_c \int_{\mathbb{R}^3} \phi_c^P(g^{-1}\mathbf{x}; G^P, \mathbf{X}^P) \phi_c^L(g^{-1}\mathbf{x}; G^L, \mathbf{X}^L) \, d^3\mathbf{x} \\
&= \sum_c \int_{\mathbb{R}^3} \phi_c^P(\mathbf{x}'; G^P, \mathbf{X}^P) \phi_c^L(\mathbf{x}'; G^L, \mathbf{X}^L) \, d^3\mathbf{x}'
\end{aligned}
$$

where the last line has substitution $\mathbf{x}' = g^{-1}\mathbf{x}$ with $g$ volume preserving on $\mathbb{R}^3$. □

## B.2 Derivations

In this section we describe the derivations for the various equations presented in the main text. We use the following convention for the (one-dimensional) Fourier transform and its inverse:

$$
\mathcal{F}[f](k) = \frac{1}{\sqrt{2\pi}} \int e^{-ikx} f(x) \, dx \tag{15a}
$$

$$
\mathcal{F}^{-1}[f](x) = \frac{1}{\sqrt{2\pi}} \int e^{ikx} f(k) \, dk \tag{15b}
$$

**Equation 5** It is well known (Wikipedia, 2023) that given a function over $\mathbb{R}^3$ with complex spherical harmonic expansion

$$
f(\mathbf{r}) = \sum_{\ell,m} f_{\ell,m}(\|\mathbf{r}\|) Y_\ell^m(\mathbf{r}/\|\mathbf{r}\|) \tag{16}
$$

its Fourier transform is given by

$$
f(\mathbf{k}) = \sum_{\ell,m} (-i)^\ell F_{\ell,m}(\|\mathbf{k}\|) Y_\ell^m(\mathbf{k}/\|\mathbf{k}\|) \tag{17}
$$

where

$$
F_{\ell,m}(k) = \frac{1}{\sqrt{k}} \int_0^\infty \sqrt{r} f_{\ell,m}(r) J_{\ell+1/2}(kr) \, r \, dr \tag{18}
$$

with $J_\ell$ the $\ell^{\text{th}}$-order Bessel function of the first kind. Relating these to the spherical Bessel functions $j_\ell$ via $J_{\ell+1/2}(x) = \sqrt{2x/\pi} j_\ell(x)$, we obtain

$$
F_{\ell,m}(k) = \sqrt{\frac{2}{\pi}} \int_0^\infty f_{\ell,m}(r) j_\ell(kr) \, r^2 \, dr \quad := \mathcal{H}_\ell[f_{\ell,m}](k) \tag{19}
$$

which is the form of Equation 6. To apply this to our scalar fields, we define the *translation operator* $\mathcal{T}_\mathbf{r}[f](\mathbf{x}) = f(\mathbf{x} - \mathbf{r})$ and note its composition with the Fourier transform

$$
(\mathcal{F} \circ \mathcal{T}_\mathbf{r})[f] = e^{-i\mathbf{k}\cdot\mathbf{r}} \mathcal{F}[f] \tag{20}
$$

We then decompose the form of our scalar fields (Equation 2) into contributions from zero-origin spherical harmonic expansions

$$\phi_c(\mathbf{x}) = \sum_n \mathcal{T}_{\mathbf{x}_n}[\phi_{cn}](\mathbf{x}) \tag{21a}$$

$$\phi_{cn}(\mathbf{x}) = \sum_{\ell,m} \underbrace{\sum_j A_{cnj\ell m} R_j(\|\mathbf{x}\|)}_{\phi_{cn\ell m}(\|\mathbf{x}\|)} Y_\ell^m(\mathbf{x}/\|\mathbf{x}\|) \tag{21b}$$

Hence, the Fourier transform of each contribution is

$$\mathcal{F}[\phi_{cn}](\mathbf{k}/\|\mathbf{k}\|) = \sum_{\ell,m} (-i)^\ell \mathcal{H}_\ell[\phi_{cn\ell m}](\|\mathbf{k}\|) Y_\ell^m(\mathbf{k}/\|\mathbf{k}\|) \tag{22}$$

Equation 5 is then obtained via Equation 20 and the linearity of the Fourier and spherical Bessel transforms.

**Equation 9**   We source (with some modifications) the derivation from Kovacs & Wriggers (2002). We consider the cross-correlation

$$c(R) = \int_{\mathbb{R}^3} \phi(\mathbf{x}) \overline{\psi(R^{-1}\mathbf{x})}\, d^3\mathbf{x} \tag{23}$$

which is the same as Equation 8 with $\phi = \phi_c^P$ and $\psi = \phi_c^L$ since $\phi_c^L$ is a real field. Expanding in complex spherical harmonics $Y_\ell^m$ and radial bases $S_j$:

$$\phi(\mathbf{x}) = \sum_{j,\ell,m} \Phi_{j\ell m} S_j(\|\mathbf{x}\|) Y_\ell^m(\mathbf{x}/\|\mathbf{x}\|) \qquad \psi(\mathbf{x}) = \sum_{j,\ell,m} \Psi_{j\ell m} S_j(\|\mathbf{x}\|) Y_\ell^m(\mathbf{x}/\|\mathbf{x}\|) \tag{24}$$

We then obtain

$$c(R) = \sum_{j,j',\ell,\ell',m,n,m'} \overline{D_{nm'}^\ell(R)} \Phi_{j\ell m} \overline{\Psi_{j'\ell'm'}} \int_{\mathbb{R}^3} [S_j \cdot S_{j'}](\|\mathbf{x}\|)[Y_\ell^m \cdot \overline{Y_{\ell'}^n}](\mathbf{x}/\|\mathbf{x}\|)\, d^3\mathbf{x} \tag{25a}$$

$$= \sum_{j,j',\ell,\ell',m,n,m'} \overline{D_{nm'}^\ell(R)} \Phi_{j\ell m} \overline{\Psi_{j'\ell'm'}} \underbrace{\int_0^\infty [S_j \cdot S_{j'}](r)\, r^2\, dr}_{G_{jj'}} \underbrace{\int_{S^2} [Y_\ell^m \cdot \overline{Y_{\ell'}^n}](\hat{\mathbf{r}})\, d\hat{\mathbf{r}}}_{\delta_{\ell\ell'}\delta_{mn}} \tag{25b}$$

$$= \sum_{\ell,m,m'} \overline{D_{mm'}^\ell(R)} \underbrace{\sum_{j,j'} \Phi_{j\ell m} \overline{\Psi_{j'\ell m'}} G_{jj'}}_{I_{mm'}^\ell} \tag{25c}$$

Now to evaluate the complex Wigner $D$-matrix, we adopt the extrinsic $zyz$ convention for Euler angles (applied right-to-left) and note that any rotation $(\phi, \theta, \psi)$ can be decomposed as

$$R(\phi, \theta, \psi) = R_z(\underbrace{\phi - \pi/2}_{\xi}) R_y(\pi/2) R_z(\underbrace{\pi - \theta}_{\eta}) R_y(\pi/2) R_z(\underbrace{\psi - \pi/2}_{\omega}) \tag{26}$$

Next, one can easily check (using the standard spherical harmonics) that the Wigner $D$-matrix for a rotation about the $z$-axis is diagonal and given by $D_{mn}^\ell(R_z(\omega)) = \delta_{mn} e^{-in\omega}$. Hence,

$$D_{mn}^\ell(R(\phi, \theta, \psi)) = e^{-im\xi} d_{mh}^\ell e^{-h\eta} d_{hn}^\ell e^{-i\omega n} \tag{27}$$

where $d^\ell = D^\ell(R_y(\pi/2))$ are constant and real. Complex conjugation then gives Equation 9.

**Equation 12**   The conditional likelihood is

$$\log p(\mathbf{t} \mid \mathbf{X}^C, R) = \log \frac{p(\mathbf{X}^C, R, \mathbf{t})}{p(\mathbf{X}^C, R)} \tag{28a}$$

$$= \log p(\mathbf{X}^C, R, \mathbf{t}) - \log \int_{\mathbb{R}^3} p(\mathbf{X}^C, R, \mathbf{t}')\, d^3\mathbf{t}' \tag{28b}$$

$$= \log E(\mathbf{X}^P, \mathbf{X}^L) - \log \int_{\mathbb{R}^3} \exp\left[E(\mathbf{X}^P, R.\mathbf{X}^C + \mathbf{t}')\right] d^3\mathbf{t}' \tag{28c}$$

Similarly,

$$\log p(R \mid \mathbf{X}^C, \mathbf{t}) = \log \frac{p(\mathbf{X}^C, R, \mathbf{t})}{p(\mathbf{X}^C, \mathbf{t})} \tag{29a}$$

$$= \log p(\mathbf{X}^C, R, \mathbf{t}) - \log \int_{SO(3)} p(\mathbf{X}^C, R', \mathbf{t}) \, dR' \tag{29b}$$

$$= \log E(\mathbf{X}^P, \mathbf{X}^L) - \log \int_{SO(3)} \exp\left[ E(\mathbf{X}^P, R.\mathbf{X}^C + \mathbf{t}) \right] \, dR' \tag{29c}$$

Finally, we move $\mathbf{t}$ to the protein coordinates (invoking the invariance of the score $E$) to obtain a form consistent with the rotational cross-correlations (Equation 8).

**Equation 13** Given a pose $\mathbf{X}^L = R.\mathbf{X}^C + \mathbf{t}$, we evaluate

$$E(\mathbf{X}^P, R.\mathbf{X}^C + \mathbf{t}) = \sum_c \int_{\mathbb{R}^3} \phi_c^P(\mathbf{x}) \phi_c^L(\mathbf{x}; R.\mathbf{X}^C + \mathbf{t}) \, d^3\mathbf{x} \tag{30}$$

The functional inner product is equivalent in Fourier space:

$$E(\mathbf{X}^P, R.\mathbf{X}^C + \mathbf{t}) = \sum_c \int_{\mathbb{R}^3} \overline{\mathcal{F}[\phi_c^P](\mathbf{k})} \cdot \mathcal{F}[\phi_c^L(\,\cdot\,; R.\mathbf{X}^C + \mathbf{t})](\mathbf{k}) \, d^3\mathbf{k} \tag{31}$$

Then with the translation operator $\mathcal{T}$ defined previously,

$$\phi_c^L(\mathbf{x}; R.\mathbf{X}^C + \mathbf{t}) = \mathcal{T}_{\mathbf{t}}[\phi(\,\cdot\,; R.\mathbf{X}^C)](\mathbf{x}) \tag{32a}$$

$$\mathcal{F}[\phi_c^L(\,\cdot\,; R.\mathbf{X}^C + \mathbf{t})](\mathbf{k}) = e^{-i\mathbf{k}\cdot\mathbf{t}} \mathcal{F}[\phi_c^L(\,\cdot\,; R.\mathbf{X}^C)](\mathbf{k}) \tag{32b}$$

We then substitute into Equation 31 to obtain Equation 13.

**Equation 14** Given a pose $\mathbf{X}^L = R.\mathbf{X}^C + \mathbf{t}$, we assume that the field $\phi_c^P(\,\cdot\,; \mathbf{X}^P - \mathbf{t})$ and $\phi_c^L(\,\cdot\,; \mathbf{X}^C)$ are written in the real global spherical harmonic expansion:

$$\phi_c^P(\mathbf{x}; \mathbf{X}^P - \mathbf{t}) = \sum_{j,\ell,m} B_{cj\ell m}^P S_j(\|\mathbf{x}\|) Y_\ell^m(\mathbf{x}/\|\mathbf{x}\|) \tag{33a}$$

$$\phi_c^L(\mathbf{x}; \mathbf{X}^C) = \sum_{j,\ell,m} B_{cj\ell m}^L S_j(\|\mathbf{x}\|) Y_\ell^m(\mathbf{x}/\|\mathbf{x}\|) \tag{33b}$$

Then, analogously to Equation 25,

$$E(\mathbf{X}^P, R.\mathbf{X}^C + \mathbf{t}) = E(\mathbf{X}^P - \mathbf{t}, R.\mathbf{X}^C) \tag{34a}$$

$$= \sum_c \int_{\mathbb{R}^3} \phi_c^P(\mathbf{x}; \mathbf{X}^P - \mathbf{t}) \phi_c^L(R^{-1}\mathbf{x}; \mathbf{X}^C) \, d^3\mathbf{x} \tag{34b}$$

$$= \sum_{c,\ell,m,m'} D_{mm'}^\ell(R) \sum_{j,j'} B_{cj\ell m}^P B_{cj'\ell m'}^L G_{jj'} \tag{34c}$$

Complex conjugation has been omitted because the coefficients and $D$-functions are now real.

# C Algorithmic Details

Below, we present in detail the four inference procedures introduced in Section 2.4. The three blocks of computations are color-coded corresponding to protein preprocessing (green), ligand preprocessing (blue), and the core computation (red) and labelled with typical runtimes from Table 1 (unlabelled lines have negligible runtime). The various loop levels make clear that depending on the workflow, the protein and ligand processing precomputations can be amortized and approaches a negligible fraction of the total runtime. Note, however, that for readability we have presented the algorithms assuming that all possible combinations (i.e., of proteins, ligand conformers, rotations, and translations) are of interest; if this is not true (for example in PDBBind, or in any typical pose-scoring setting), then the full benefits of amortization may not be fully realized.

---

**Algorithm 1:** TRANSLATIONAL FFT

---

**Input:** Proteins $\{(G_i^P, \mathbf{X}_i^P)\}$, conformers $\{(G_h^L, \mathbf{X}_h^L)\}$
**Output:** Docked poses $(\mathbf{X}_i^P, \mathbf{X}_{ih}^L) \, \forall i, h$

**foreach** $(G_i^P, \mathbf{X}_i^P)$ **do**                    // protein preprocessing
  Compute coefficients $\mathbf{A}_i^P = \{A_{cjn\ell m}^P(G_i^P, \mathbf{X}_i^P)\}$ with neural network ;        // 65 ms
  Compute Fourier-space field values $\mathcal{F}[\phi^P]_i$ using $\mathbf{A}_i^P, \mathbf{x}_i^P$ ;        // 7.0 ms

**foreach** $(G_h^L, \mathbf{X}_h^L)$ **do**                    // ligand preprocessing
  Compute coefficients $\mathbf{A}_h^L = \{A_{cjn\ell m}^L(G_h^L, \mathbf{X}_h^L)\}$ with neural network ;        // 4.3 ms
  **foreach** $R_k \in \{R\}_{grid} \subset SO(3)$ **do**
    Compute rotated coefficients $\mathbf{A}_{h,k}^L$ using $D^\ell(R_k)$;
    Compute Fourier-space field values $\mathcal{F}[\phi^L]_{h,k}$ using $\mathbf{A}_{h,k}^L, R_k\mathbf{X}_h^L$ ;        // 1.6 ms

**foreach** $(G_i^P, \mathbf{X}_i^P)$ **do**                    // pose optimization
  **foreach** $(G_h^L, \mathbf{X}_h^L)$ **do**
    **foreach** $R_k \in \{R\}_{grid} \subset SO(3)$ **do**
      Compute $E(\mathbf{X}_i^P, R_k\mathbf{X}_h^L + \mathbf{t}), \forall \mathbf{t}$ using FFT;        // 160 $\mu$s
      $E_k^\star, \mathbf{t}_k^\star \leftarrow \{\max, \arg\max\}_\mathbf{t} E(\mathbf{X}_i^P, R_k\mathbf{X}_h^L + \mathbf{t})$ ;
    $k^\star \leftarrow \arg\max_k E_k^\star$;
    $\mathbf{X}_{ih}^L \leftarrow R_{k^\star}\mathbf{X}_h^L + \mathbf{t}_{k^\star}^\star$;

---

**Algorithm 2:** ROTATIONAL FFT

---

**Input:** Proteins $\{(G_i^P, \mathbf{X}_i^P)\}$, conformers $\{(G_h^L, \mathbf{X}_h^L)\}$
**Output:** Docked poses $(\mathbf{X}_i^P, \mathbf{X}_{ih}^L) \, \forall i, h$

**foreach** $(G_i^P, \mathbf{X}_i^P)$ **do**                    // protein preprocessing
  Compute coefficients $\mathbf{A}_i^P = \{A_{cjn\ell m}^P(G_i^P, \mathbf{X}_i^P)\}$ with neural network ;        // 65 ms
  **for** $\mathbf{t}_k \in \{\mathbf{t}\}_{grid} \subset \mathbb{R}^3$ **do**
    Compute global expansion $\mathbf{B}_{i,k}^P = \{B_{cj\ell m}\}$ from $\mathbf{A}_i^P, \mathbf{X}_i^P - \mathbf{t}_k$ ;        // 80 ms

**foreach** $(G_h^L, \mathbf{X}_h^L)$ **do**                    // ligand preprocessing
  Compute coefficients $\mathbf{A}_h^L = \{A_{cjn\ell m}^L(G_h^L, \mathbf{X}_h^L)\}$ with neural network ;        // 4.3 ms
  Compute global expansion $\mathbf{B}_h^L = \{B_{cj\ell m}\}$ from $\mathbf{A}_h^L, \mathbf{X}_h^L$ ;        // 17 ms

**foreach** $(G_i^P, \mathbf{X}_i^P)$ **do**                    // pose optimization
  **foreach** $(G_h^L, \mathbf{X}_h^L)$ **do**
    **foreach** $\mathbf{t}_k \in \{\mathbf{t}\}_{grid} \subset \mathbb{R}^3$ **do**
      Compute $E(\mathbf{X}_i^P - \mathbf{t}_k, R.\mathbf{X}_h^L), \forall R$ using FFT ;        // 650 $\mu$s
      $E_k^\star, R_k^\star \leftarrow \{\max, \arg\max\}_R E(\mathbf{X}_i^P - \mathbf{t}_k, R.\mathbf{X}_h^L + \mathbf{t})$ ;
    $k^\star \leftarrow \arg\max_k E_k^\star$;
    $\mathbf{X}_{ih}^L \leftarrow R_{k^\star}^\star\mathbf{X}_h^L + \mathbf{t}_{k^\star}$;

**Algorithm 3:** TRANSLATIONAL SCORING

**Input:** Proteins $\{(G_i^P, \mathbf{X}_i^P)\}$, conformers $\{(G_h^L, \mathbf{X}_h^L)\}$, rotations $\{R_k\}$, translations $\{\mathbf{t}_\ell\}$
**Output:** Scores $E(\mathbf{X}_i^P, R_k\mathbf{X}_h^L + \mathbf{t}_\ell)\ \forall i, h, k, \ell$

**foreach** $(G_i^P, \mathbf{X}_i^P)$ **do**                                                   // protein preprocessing
   Compute coefficients $\mathbf{A}_i^P = \{A_{cjn\ell m}^P(G_i^P, \mathbf{X}_i^P)\}$ with neural network ;        // 65 ms
   Compute Fourier-space field values $\mathcal{F}[\phi^P]_i$ using $\mathbf{A}_i^P, \mathbf{x}_i^P$ ;        // 7.0 ms

**foreach** $(G_h^L, \mathbf{X}_h^L)$ **do**                                                   // ligand preprocessing
   Compute coefficients $\mathbf{A}_h^L = \{A_{cjn\ell m}^L(G_h^L, \mathbf{X}_h^L)\}$ with neural network ;        // 4.3 ms
   **foreach** $R_k$ **do**
      Compute rotated coefficients $\mathbf{A}_{h,k}^L$ using $D^\ell(R_k)$;
      Compute Fourier-space field values $\mathcal{F}[\phi^L]_{h,k}$ using $\mathbf{A}_{h,k}^L, R_k\mathbf{X}_h^L$ ;        // 1.6 ms

**foreach** $(G_i^P, \mathbf{X}_i^P)$ **do**                                                   // scoring
   **foreach** $(G_h^L, \mathbf{X}_h^L)$ **do**
      **foreach** $R_k$ **do**
         **foreach** $\mathbf{t}_\ell$ **do**
            Compute $E(\mathbf{X}_i^P, R_k\mathbf{X}_h^L + \mathbf{t}_\ell)$ using Equation 13;        // 1.0 $\mu$s

---

**Algorithm 4:** ROTATIONAL SCORING

**Input:** Proteins $\{(G_i^P, \mathbf{X}_i^P)\}$, conformers $\{(G_h^L, \mathbf{X}_h^L)\}$, rotations $\{R_k\}$, translations $\{\mathbf{t}_\ell\}$
**Output:** Scores $E(\mathbf{X}_i^P, R_k\mathbf{X}_h^L + \mathbf{t}_\ell)\ \forall i, h, k, \ell$

**foreach** $(G_i^P, \mathbf{X}_i^P)$ **do**                                                   // protein preprocessing
   Compute coefficients $\mathbf{A}_i^P = \{A_{cjn\ell m}^P(G_i^P, \mathbf{X}_i^P)\}$ with neural network ;        // 65 ms
   **for** $\mathbf{t}_k \in \{\mathbf{t}\}_{grid} \subset \mathbb{R}^3$ **do**
      Compute global expansion $\mathbf{B}_{i,k}^P = \{B_{cj\ell m}\}$ from $\mathbf{A}_i^P, \mathbf{X}_i^P - \mathbf{t}_k$ ;        // 80 ms

**foreach** $(G_h^L, \mathbf{X}_h^L)$ **do**                                                   // ligand preprocessing
   Compute coefficients $\mathbf{A}_h^L = \{A_{cjn\ell m}^L(G_h^L, \mathbf{X}_h^L)\}$ with neural network ;        // 4.3 ms
   Compute global expansion $\mathbf{B}_h^L = \{B_{cj\ell m}\}$ from $\mathbf{A}_h^L, \mathbf{X}_h^L$ ;        // 17 ms

**foreach** $(G_i^P, \mathbf{X}_i^P)$ **do**                                                   // scoring
   **foreach** $(G_h^L, \mathbf{X}_h^L)$ **do**
      **foreach** $R_k$ **do**
         **foreach** $\mathbf{t}_\ell$ **do**
            Compute $E(\mathbf{X}_i^P, R_k\mathbf{X}_h^L + \mathbf{t}_\ell)$ using Equation 14;        // 8.2 $\mu$s

# D    Learned Scalar Fields

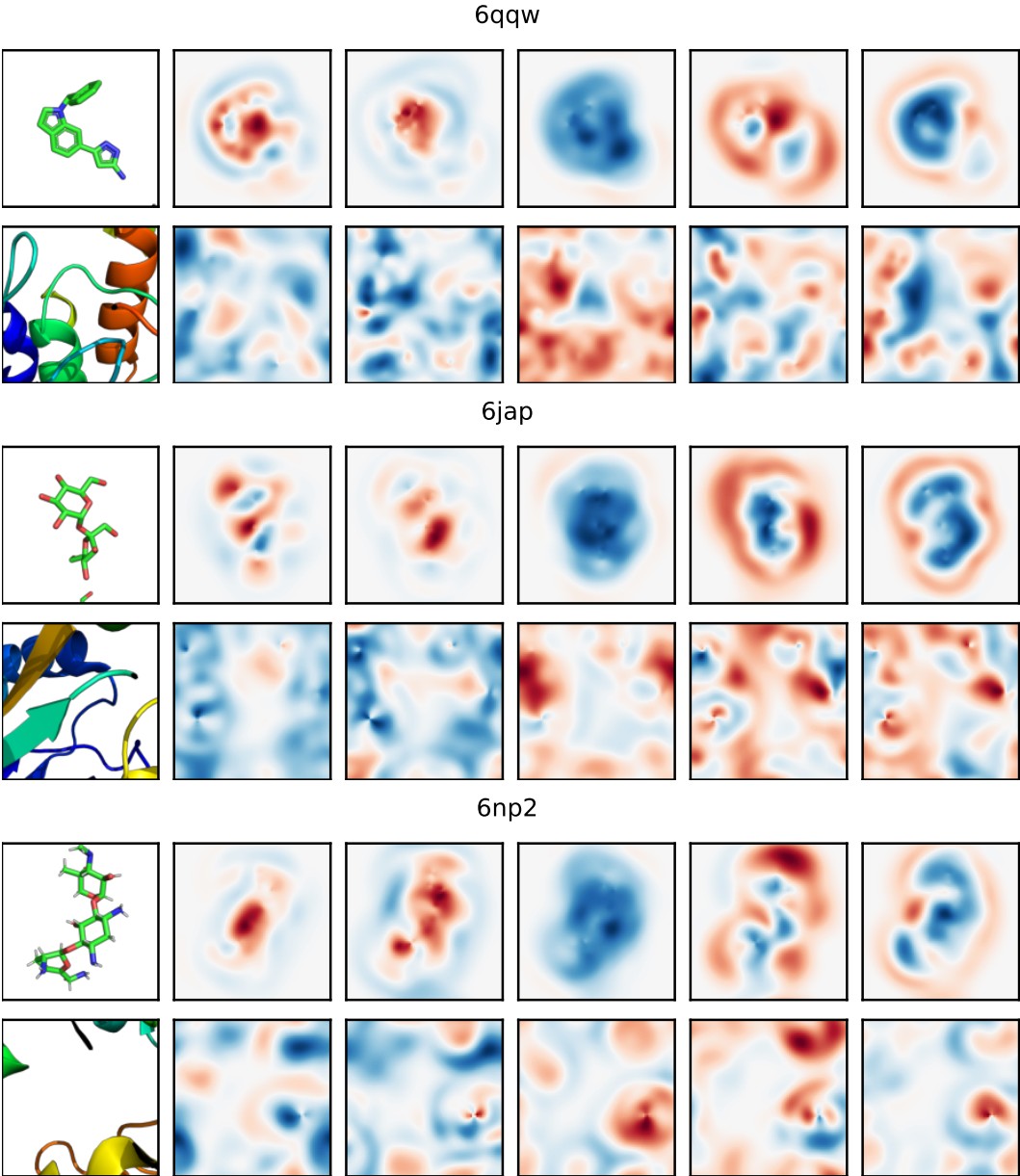

Figure 2: **Visualizations of learned scalar fields**. All five channels of the **ESF-N** learned scalar fields $\phi^L$ (top row) and $\phi^P$ (bottom row) are shown on the $xy$-plane passing through the center of mass of the ligand, with a box diameter of 20 Å. Positive values of the field are in blue and negative values in red. At left, the ligand and pocket structures are shown looking down the $z$-axis. Note that as the fields are only 2D slices, not all 3D features visible in the structures are visible in the fields.

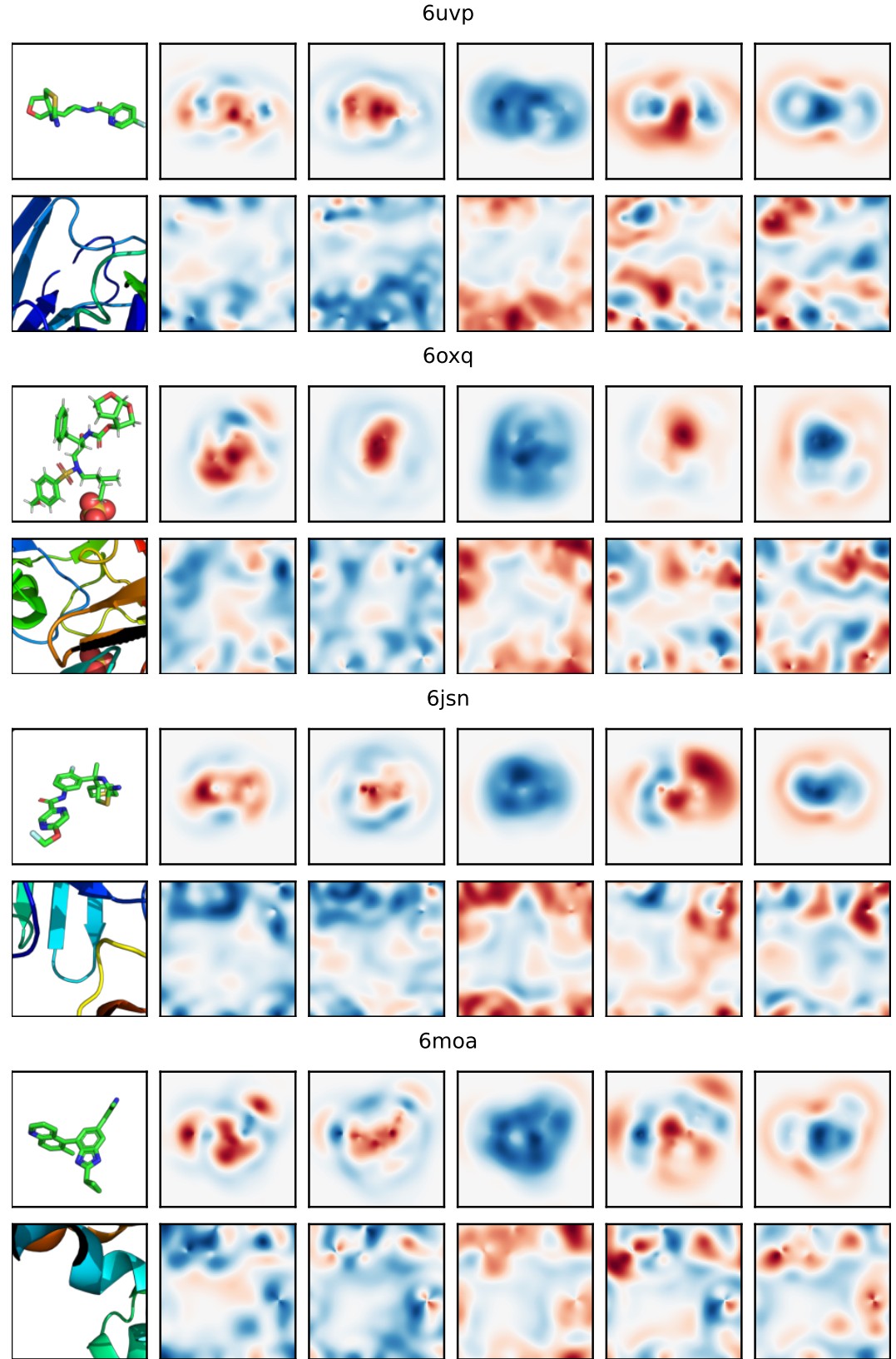

Figure 3: **Visualizations of learned scalar fields**, continued.

# E   Experimental Details

## E.1   Decoy Set

Given a zero-mean ground-truth ligand pose $\mathbf{X}^{L^\star}$, we generate $32^3 - 1 = 32767$ decoy poses via the following procedure.

- Sample 31 translational pertubations: $\mathbf{t}_i \sim \mathcal{N}(0, \mathbf{I}_3), i = 1 \ldots 31$ and set $\mathbf{t}_0 = \mathbf{0}$, with units in Å.
- Sample 31 rotational perturbations: $R_j = \texttt{FromRotvec}(\mathbf{r}_j), \mathbf{r}_j \sim \mathcal{N}(0, 0.5\mathbf{I}_3), j = 1 \ldots 31$ and set $R_0 = \mathbf{I}_3$.
- Sample 31 noisy conformers $\mathbf{X}_k^C, k = 1 \ldots 31$ by sampling torsional updates $\Delta \boldsymbol{\tau}_k \sim \mathcal{N}_\mathbb{T}(0, (\pi/2)\mathbf{I}_m)$ where $\mathcal{N}_\mathbb{T}$ is a wrapped normal distribution (Jing et al., 2022) and $m$ is the number of torsion angles. The torsional updates are applied to the smaller side of the molecule. Set $\mathbf{X}_0^C = \mathbf{X}^{L^\star}$.
- Set $\mathbf{X}_{ijk}^L = R_j \mathbf{X}_k^C + \mathbf{t}_i, i, j, k = 0 \ldots 31$ and discard $\mathbf{X}_{000}^L = \mathbf{X}^{L^\star}$.

PDB ID 6A73 is excluded from the procedure due to the high level of graph symmetry and significant runtime for computing RMSDs for all decoys. Summary statistics for the decoy sets of the remaining 362 PDB IDs are presented in Figure 4.

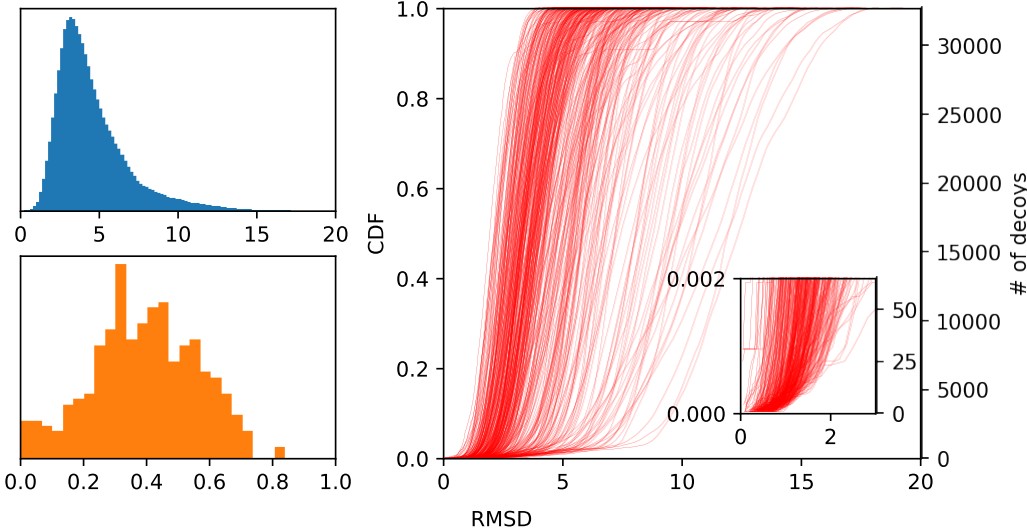

Figure 4: **Decoy set statistics.** *Top left*: histogram of RMSDs across all decoys sets (12 M total). *Bottom left*: histogram of minimum RMSDs among the decoy sets. All sets have a pose less than RMSD <1 Å from the true pose. *Right*: cumulative density function of RMSDs in each decoy set. *Bottom right inset*: all decoy sets have at least 23 poses with RMSD <2 Å.

## E.2   Hyperparameters

Our method involves hyperparameters at several levels.

- The learned scalar fields have 5 channels.
- To parameterize the scalar field (Equation 2), we use spherical harmonics up to $\ell = 2$ and 5 Gaussian RBFs evenly spaced from 0 Å to 5 Å.
- All translational Fourier coefficients (Equation 5) are evaluated with a grid of frequencies corresponding to a sampling interval of 1 Å and a cubical domain with side length 40 Å. The integral over $\mathbb{R}^3$ in the training objective (Equation 12) is computed only over the

cubical FFT domain. During training and default inference, the cross-correlation is also computed with a sampling interval of 1 Å, but denser sampling intervals at inference-time (i.e., by zero-padding in the Fourier domain) are explored in Appendix F.

- Global spherical harmonic expansions (Equation 7) are computed up to $\ell = 10$, with 25 Gaussian RBFs evenly spaced from 0 Å to 20 Å. During training and default inference, the evaluation of rotational cross-correlations with FFTs (Equation 9) is always performed up to $\ell = 50$ and $\ell = 25$, respectively, by zero-padding in the Fourier domain, with other inference orders at inference-time explored in Appendix F.

- Local-to-global transformation matrices (Equation 11) were precomputed for discretized positions along the $+z$ axis from 0 Å to 20 Å at 1 Å intervals.

- Data featurization and model hyperparameters are adapted from the default settings of Corso et al. (2023), giving a model size of 2.2 M parameters for both the ligand and protein model.

- By default, in the **RF** procedure, we evaluate $9^3 = 729$ translational grid points at inference time, filling a 8 Å cube at 1 Å intervals. In the **TF** procedure, we use an $m = 2$ grid over $SO(3)$ as implemented by Zhong et al. (2019) and Yershova et al. (2010), yielding 4608 grid points. Other resolutions are explored in Appendix F.

These hyperparameters were not extensively tuned, and further tradeoffs and improvements in performance and runtime could be explored by modifying them.

### E.3 Runtime Measurements

All runtime measurements were performed on a machine with 64 Intel Xeon Gold 6130 CPUs and 8 Nvidia Tesla V100 GPUs. Gnina was run with default thread count settings. All of our processes were run on a single V100 GPU. For our method, we performed runtime analysis using CUDA events to remove the effects of asynchronous CUDA execution. Script loading, model loading, and algorithmic-level precomputations (which, if necessary, can be cached on disk) were excluded from the analysis. For Gnina, we attempted to remove similar overhead by timing single-pose scoring-only runs as representative of constant overhead costs. We report conformer docking runtimes in Table 3 using the PDBBind crystal structures; ESMFold runtimes are marginally shorter. Typical runtimes reported in Table 1 and Appendix C are obtained from timing runs with our method across the entire PDBBind test set.

### E.4 Datasets

As noted previously, we use train, validation, and test splits from Stärk et al. (2022). However, due to RDKit parsing issues with Gnina-docked poses, the following 30 complexes are excluded (leaving 333 remaining) from all rigid conformer docking comparisons against Gnina, i.e., Tables 3 and Appendix F: 6HZB, 6E4C, 6PKA, 6E3P, 6OXT, 6OY0, 6HZA, 6E6W, 6OXX, 6HZD, 6K05, 6NRH, 6OXW, 6RTN, 6D3Z, 6HLE, 6PY0, 6OXS, 6E3O, 6HZC, 6Q38, 6E7M, 6OIE, 6D3Y, 6D40, 6UHU, 6CJP, 6E3N, 6Q4Q, 6D3X. Scoring comparisons include all test complexes except 6A73, for which decoy poses could not be generated.

We download the 77 PDB IDs provided in Tosstorff et al. (2022) from the PDB to form the PDE10A dataset, keeping the A chain of each assymetric unit and the Ligand of Interest (LOI) interacting with it. We then align all ligands to the crystal structure of 5SFS using the procedure described in Corso et al. (2023) for aligning ESMFold structures, except transforming the ligand rather than the protein. This constitutes the construction of a *cross-docking* dataset due to the use of the same pocket for all ligands. Due to RDKit parsing errors with the Gnina-docked poses, the following 7 PDB IDs are excluded from all comparisons: 5SFA, 5SED, 5SFO, 5SEV, 5SF9, 5SDX, 5SFC. The remaining 70 ligands are shown superimposed on the 5SFS pocket in Figure 5.

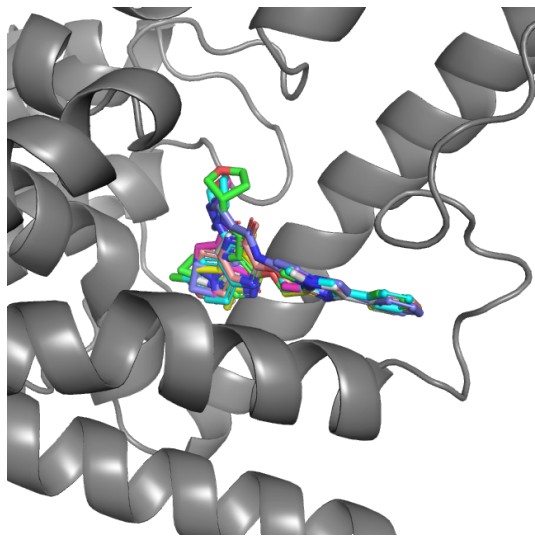

Figure 5: **PDE10A ligands** aligned on 5SFS

# F  Further Results

In Tables 4–7 below, we explore the impact of inference-time hyperparameters on the performance and runtime of our method on the rigid conformer docking task. We use the **ESF-N** model variant and experiment with the PDBBind crystal test set and PDE10A test set. For the **TF** procedure (Algorithm 1), we adjust (1) the number of grid points over $SO(3)$ with two possible resolutions, giving 576 and 4608 possible rotations, respectively; (2) the spatial interval at which the translational cross-correlation (Equation 3) is evaluated: either 1 Å or 0.5 Å (by zero-padding the scalar fields in the Fourier domain), giving a cubical grid of evaluated points with side length 8 Å and 9 or 17 points on each side. For the **RF** procedure (Algorithm 2), we adjust (1) the number of translational grid points with three possible resolutions, filling the 8 Å cube with 7, 9, or 13 points per side, respectively; (2) the resolution at which the rotational cross-correlation (Equation 8) is evaluated; this can be adjusted by zero-padding in the Fourier domain to include larger values of $\ell$. The rows corresponding to results in the main Table 3 are bolded.

In all rows, the effective number of poses searched over via both degrees of freedom is computed. To provide an idea of the impact of discretization, we compute the median RMSD of the *closest grid point* to the ground-truth pose (decomposed into rotational and translational contributions). This serves as a hard lower bound for the median RMSD of the output docked pose. In the **TF** procedure, increasing the resolution is memory-intensive; thus, the **RF** procedure is more effective at leveraging FFT to conduct fine-grained search over the accelerated degree of freedom. The default reported performance is attained with a translational offset of 0.4 Å and a rotational offset of 0.16 Å. While performance improves with smaller grid offsets, the returns are rapidly dimishing.

The runtime of the method (averaged over 333 PDBBind complexes and 70 PDE10A complexes) is reported and color-coded according to Appendix C: protein preprocessing (green), ligand preprocessing (blue), and the pose optimization (red). The effect of the non-FFT grid resolution is also color-coded, i.e., in **TF** the explicit enumeration over $SO(3)$ grid points directly scales the ligand preprocessing, whereas in **RF** the enumeration over $\mathbb{R}^3$ scales the protein preprocessing. As the tables show, the preprocessing of these explicit grid points contributes to the majority of the non-amortizeable runtime. In general, the $SO(3)$ grid / ligand preprocessing in **TF** is less expensive, however, it cannot be amortized when moving from PDBBind to PDE10A (where the ligands are still distinct). On the other hand, the $\mathbb{R}^3$ grid / protein preprocessing time in **RF** is significantly reduced (very roughly on the order of 70-fold, as expected) in PDE10A compared to PDBBind.

Table 4: **PDBBind TF**

| Trans. grid | $SO(3)$ grid | Effective # poses | Grid offset Tr. | Rot. | All | Med. RMSD | % <2Å | Prot. prep. | Lig. prep. | Opt. |
|---|---|---|---|---|---|---|---|---|---|---|
| 9 | 576 | 420k | 0.52 | 0.84 | 0.98 | 1.53 | 63 | 65 | 931 | 100 |
| **9** | **4608** | **3.4M** | **0.50** | **0.42** | **0.67** | **1.10** | **72** | **72** | **7196** | **715** |
| 17 | 576 | 2.8M | 0.25 | 0.80 | 0.84 | 1.50 | 64 | 70 | 928 | 123 |

Table 5: **PDBBind RF**

| Trans. grid | $SO(3)$ $\ell_{max}$ | Effective # poses | Grid offset Tr. | Rot. | All | Med. RMSD | % <2Å | Prot. prep. | Lig. prep. | Opt. |
|---|---|---|---|---|---|---|---|---|---|---|
| 7 | 10 | 3.2M | 0.65 | 0.38 | 0.80 | 1.25 | 70 | 30k | 85 | 158 |
| 7 | 25 | 45M | 0.67 | 0.15 | 0.70 | 1.15 | 69 | 31k | 87 | 225 |
| 7 | 50 | 353M | 0.65 | 0.08 | 0.67 | 1.16 | 70 | 32k | 85 | 704 |
| 9 | 10 | 6.8M | 0.49 | 0.36 | 0.64 | 1.16 | 73 | 64k | 85 | 333 |
| **9** | **25** | **97M** | **0.50** | **0.15** | **0.53** | **1.00** | **73** | **67k** | **87** | **476** |
| 9 | 50 | 751M | 0.51 | 0.08 | 0.52 | 0.98 | 74 | 63k | 84 | 1487 |
| 13 | 10 | 20M | 0.33 | 0.37 | 0.51 | 1.05 | 74 | 198k | 85 | 995 |
| 13 | 25 | 291M | 0.33 | 0.15 | 0.37 | 0.90 | 72 | 200k | 86 | 1430 |

Table 6: **PDE10A TF**

| Trans. grid | $SO(3)$ grid | Effective # poses | Grid offset Tr. | Rot. | All | Med. RMSD | % <2Å | Prot. prep. | Lig. prep. | Opt. |
|---|---|---|---|---|---|---|---|---|---|---|
| 9 | 576 | 420k | 0.51 | 0.88 | 1.00 | 1.85 | 56 | 22 | 761 | 89 |
| **9** | **4608** | **3.4M** | **0.50** | **0.48** | **0.69** | **1.11** | **64** | **21** | **6159** | **736** |
| 17 | 576 | 2.8M | 0.26 | 0.89 | 0.93 | 2.05 | 50 | 20 | 756 | 106 |
| 17 | 4608 | 23M | 0.26 | 0.44 | 0.51 | 1.00 | 73 | 20 | 6147 | 892 |

Table 7: **PDE10A RF**

| Trans. grid | $SO(3)$ $\ell_{max}$ | Effective # poses | Grid offset Tr. | Rot. | All | Med. RMSD | % <2Å | Prot. prep. | Lig. prep. | Opt. |
|---|---|---|---|---|---|---|---|---|---|---|
| 7 | 10 | 3.2M | 0.72 | 0.38 | 0.83 | 1.60 | 54 | 476 | 44 | 161 |
| 7 | 25 | 45M | 0.57 | 0.16 | 0.59 | 1.21 | 63 | 549 | 42 | 227 |
| 7 | 50 | 353M | 0.65 | 0.08 | 0.65 | 1.30 | 64 | 635 | 59 | 718 |
| 9 | 10 | 6.8M | 0.46 | 0.39 | 0.63 | 1.05 | 64 | 1014 | 42 | 327 |
| **9** | **25** | **97M** | **0.48** | **0.16** | **0.51** | **1.00** | **70** | **946** | **43** | **465** |
| 9 | 50 | 751M | 0.49 | 0.09 | 0.50 | 0.99 | 64 | 943 | 42 | 1483 |
| 13 | 10 | 20M | 0.34 | 0.41 | 0.55 | 1.17 | 64 | 2798 | 42 | 986 |
| 13 | 25 | 291M | 0.33 | 0.16 | 0.36 | 0.96 | 69 | 2912 | 45 | 1469 |

In Figure 6, we further investigate the tradeoff between speed and performance offered by our method compared to Gnina (with the Vina scoring function). While in the main results (Table 3) we run Gnina using all default settings, it is possible to reduce the runtime (and performance) by adjusting these settings. In particular, we explore setting `--max_mc_steps` and `--minimize_iters` to 5 independently and in combination. Together with the default runs and the `--score_only` runs, these trace out a *Pareto frontier* representing the tradeoff between runtime per complex and <2 Å RMSD success rate. With the default settings, Gnina outperforms all variants of our method on the PDBBind crystal and PDE10A test sets. However, Figure 6 shows that we can reach previously inaccessible regions in the accuracy v.s. runtime tradeoff landscape.

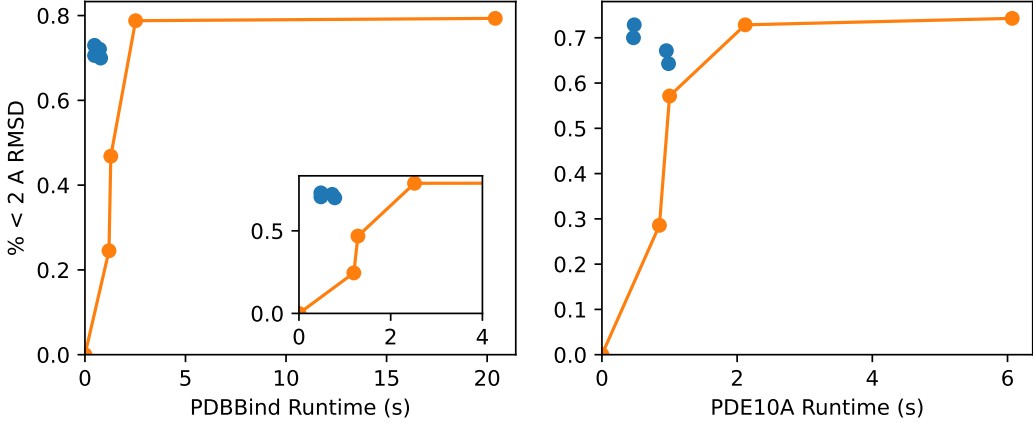

Figure 6: **Tradeoff between speed and accuracy** using our method compared to Gnina on PDBBind crystal structures (*left*) and PDE10A (*right*). In both cases, variants of our method (blue dots) enable possibilities not reachable with Gnina (orange curve).

