# OpenReview forum: "Learning Scalar Fields for Molecular Docking with Fast Fourier Transforms"
_NeurIPS.cc/2023/Workshop/AI4Science — NeurIPS2023-AI4Science Poster_

### Official Review · Reviewer_NKKV · 2023-10-08
**A new scoring function and optimization protocol to speed up docking tasks**

**Rating:** 8
**Confidence:** 3

**Review:**

**Overview**

The paper discusses a novel way to speed up the molecular docking process by suggesting a new scoring function that is the cross-correlation of multi-channel ligand and protein scalar fields and an optimization approach using FFT. This is a well-written and well-tested work, and I recommend to accept this work to AI for Science workshop. I feel this work also provide the community another scope to study the docking problem via ML. Here are some detailed comments for this work.
* Disclaimer: I didn't check the correctness of the equations and derivations considering the length of this paper is pretty long and evaluate the paper mostly on its motivations and results.

**Details**
* **Novelty**: Although molecular docking problem is wildly studied in both traditional computational biology and AI for science, the specific application of cross-correlation of scalar fields and FFTs in this problem is quite unique. Most of the previous studies only proposed ML-based scoring functions but largely ignore the actual computational costs in the optimizations. This particular feature helps set this paper apart from other studies in the same arena and could potentially help the community to explore the study directions in the future.
* **Writing quality**: The writing quality of this work is pretty good which makes it easy for readers to follow the argument and key points. Since AI for Science workshop relaxes the page requirements to 4-8 pages, it is ok that the length of this work is longer than a traditional workshop paper.
* **Results**: This work has already done a lot of nice experiments and most of the results are solid. I just raise one comment on what I think might be helpful.
  * Usage of Equivariant graph neural networks, uncertainty quantification & possibility of active learning: In this work, the author tests on the PDBBind dataset containing over 16 K training set and also need to sample different pose structures as indicated in the paper. To make this approach indeed useful, it is important to consider the issue of lacking of enough experimental data. It will be valuable for the authors continue working on their Equivariant GNN and considering how to produce reliable UQ and even an active learning approach to reduce the required number of training data and more efficient sampling.

---

### Official Review · Reviewer_t4if · 2023-10-23
**Interesting work, but does not follow workshop guidelines**

**Rating:** 4
**Confidence:** 4

**Review:**

I think this work covers a domain that would be interesting for the AI4Science community, and contains a lot of compelling approaches and experiments. However, the main paper is 9 full pages long, which does not conform to the 4-8 pages set by this workshop.

Generally, I think refining the conclusions of research and message for the audience is an important skill. The authors should determine what needs to be edited down and moved to the appendix to submit to this workshop, or find a different venue that can accommodate for this paper length.

---

### Meta-Review · Area_Chair_rC1i · 2023-10-26

**Recommendation:** Accept (Oral)
**Confidence:** 4

**Metareview:**

This paper proposes a novel scoring function for the molecular docking process derived from the cross-correlation of multi-channel ligand and protein scalar fields, coupled with an optimization approach using FFT. Both reviewers praise its novelty, experimental evaluations and writing quality. The paper exceeds the workshop's 8-page guideline, I'd kindly suggest the author to refine his content before publication.